# Transition Metal-Hyperdoped InP Semiconductors as Efficient Solar Absorber Materials

**DOI:** 10.3390/nano10020283

**Published:** 2020-02-07

**Authors:** Gregorio García, Pablo Sánchez-Palencia, Pablo Palacios, Perla Wahnón

**Affiliations:** 1Instituto de Energía Solar, ETSI Telecomunicación, Universidad Politécnica de Madrid, Ciudad Universitaria, s/n, 28040 Madrid, Spain; p.sanchez-palencia@upm.es (P.S.-P.); pablo.palacios@upm.es (P.P.); perla@etsit.upm.es (P.W.); 2Departamento de Tecnología Fotónica y Bioingeniería, ETSI Telecomunicación, Universidad Politécnica de Madrid, Ciudad Universitaria, s/n, 28040 Madrid, Spain; 3Departamento de Física aplicada a las Ingenierías Aeronáutica y Naval, ETSI Aeronáutica y del Espacio, Universidad Politécnica de Madrid, Pz. Cardenal Cisneros, 3, 28040 Madrid, Spain

**Keywords:** transition metal-hyperdoped, InP, photovoltaic, DFT, GW, in-gap band

## Abstract

This work explores the possibility of increasing the photovoltaic efficiency of InP semiconductors through a hyperdoping process with transition metals (TM = Ti, V, Cr, Mn). To this end, we investigated the crystal structure, electronic band and optical absorption features of TM-hyperdoped InP (TM@InP), with the formula TMxIn1-xP (x = 0.03), by using accurate ab initio electronic structure calculations. The analysis of the electronic structure shows that TM 3d-orbitals induce new states in the host semiconductor bandgap, leading to improved absorption features that cover the whole range of the sunlight spectrum. The best results are obtained for Cr@InP, which is an excellent candidate as an in-gap band (IGB) absorber material. As a result, the sunlight absorption of the material is considerably improved through new sub-bandgap transitions across the IGB. Our results provide a systematic and overall perspective about the effects of transition metal hyperdoping into the exploitation of new semiconductors as potential key materials for photovoltaic applications.

## 1. Introduction

Despite the emergence of new renewable energy technologies, solar energy is still attracting a great deal of interest due to its abundant, clean and renewable characteristics. Even though silicon solar cells dominate the market share for photovoltaic (PV) technologies, there are considerable efforts to explore other semiconductor materials that could be used in photovoltaic devices [1,2]. To be an adequate solar cell material, a semiconductor would need to have a direct bandgap (to enhance light absorption and minimize nonradiative recombination). C.H. Henry estimated the thermodynamic limit of single-energy gap photovoltaic devices, pointing out that the conversion efficiency exhibits the highest values for bandgap values at ∼1.10 and ∼1.35 eV [3], being crystalline devices closer to the thermodynamic limit. T. Unold et al. assessed the relationship between the bandgap and the maximum obtainable open-circuit voltage (which is related to nonradiative recombination) [2,3], pointing out that crystalline materials such as GaAs and InP could reach the highest open-circuit voltage values (closer to the thermodynamic limit). Both InP and GaAs are III-V semiconductor materials with a direct bandgap (1.35 and 1.42 eV for InP and GaAs, respectively) [4] close to the optimum for solar energy conversion making them suitable candidates for solar cells. As matter of fact, energy conversion efficiencies of around 40% have been obtained for III–V semiconductors in multijunction tandem architecture at the laboratory scale [5]. GaAs has been widely studied despite the high toxicity of As atom [2,5,6,7,8]. Therefore, we have focused on InP, since this semiconductor achieves an optimal combination of bandgap width along with a high crystalline quality, to efficiently convert solar radiation into electrical energy. Unfortunately, even the most efficient InP solar cells have only reached efficiencies up to 22.1% [9,10,11,12]. Consequently, its efficiency as a solar cell material must be improved to reach up to (or even over) the theoretical maximum conversion efficiency ~ 33% based on the Shockley–Queisser (SQ) model [13].

Among the alternative designs that, theoretically, can exceed the SQ limit, large efforts have been made based on the in-gap band (IGB) concept. Those materials with an IGB (also known as an intermediate band) own a partially filled narrow band located between the valence and conduction bands (VB and CB, respectively) of the host semiconductor (see Figure 1). Thus, IGB materials could improve the conversion efficiency through a three-photon absorption process: from the VB to the CB, along with the absorption of two extra sub-bandgap photons (from the VB to the IGB and, from there, to the CB). This would lead to the creation of additional electron-hole pairs and a higher photocurrent without a decrease in the open-circuit voltage. Thus, the upper limit efficiency of an IGB solar cell could reach theoretical efficiencies of up to 63.1% [14,15,16]. In addition, the IGB must meet some additional requirements to improve its photovoltaic efficiency: (i) have its own small dispersion, without it being at a discreet level; (ii) be well isolated from the VB and CB to avoid thermalization between VB and CB; (iii) be partially filled to allow comparable rates between VB–IGB and IGB–CB transitions; (iv) have high concentrations of IGB states to produce high absorption coefficients for those sub-bandgap absorptions and avoid non-radiative recombination obtained through the formation of a highly delocalized energy band [14,15,16]. Among the major approaches considered in designing IGB materials, our group have extensively studied the formation of an IGB through the substitutional doping of some cations by a transition metal (TM) [17,18,19,20,21,22,23,24,25,26,27,28,29,30,31,32,33,34,35,36]. In this way, the *d*-orbitals of the TM might be located within the bandgap of the semiconductors, allowing for the formation of an isolated energy band, while the filling of the IGB can be finetuned through the adequate selection of the TM. To our knowledge, proposals for the creation of an IGB in InP semiconductor have not been suggested. Recently, J. Olea et al. synthetized GaP supersaturated with Ti by means of ion implantation, followed by pulsed-laser melting. The spectral photoconductivity measurements yielded an enhancement in the conductivity for energies below the bandgap of GaP, which were attributed to the presence of an IGB within the GaP bandgap [37]. The formation of an IGB in GaP hyperdoped with Ti was previously reported by our group using ab initio calculations [17,38,39].

The main goal of this paper is the study of the structural and electronic properties of several TM-hyperdoped InP (TM@InP) compounds in order to find new IGB materials with improved absorption features. Concretely, the selected TM were early 3*d*-block elements, whose ability to form an IGB has been previously demonstrated [17,20,22,23,25,26,28,29,30,31,33,34,35,36,38,39,40,41,42,43,44]. Hence, in this work we investigate the structural, stability and electronic properties of several TM@InP (TM = Ti, V, Cr, Mn). It is well-known that standard density functional theory (DFT) methods are adequate to compute the structural and thermodynamic properties. The electronic properties of a semiconductor material are mainly characterized through the bandgap (*E_g_*) value. Unfortunately, the commonly used DFT exchange–correlation functionals (such as the PBE one) dramatically underestimate the bandgap [45]. Aimed at providing an accurate description of the electronic structure and optical properties, DFT methods, along with the formalism proposed by Dudarev et al. (DFT+*U*) [46], were used. This method allows for a correct description of the band structure and density of states, but keeps the simplicity and excellent quality-to-cost ratio of standard DFT methods [47]. Although PBE+*U* formalism would be adequate for the native host material, such an approach has not been applied to in-gap band materials. Our previous works have demonstrated that the many-body perturbation theory in *GW* approximation (concretely *G*_0_*W*_0_ approach) [48], yields electronic structure information according to experimental data for other transition metal semiconductors with an in-gap band [24,34,35]. As a result, the electronic structure and optical absorption features were calculated using DFT+*U*, while *G_0_W_0_* was used as benchmark method. We find that TM-hyperdoped InP semiconductors lead to an important absorption improvement, obtaining the most promising results for Cr@InP.

## 2. Computational Methodology

All calculations proposed here were performed using the Vienna ab initio simulation package (VASP) [49,50] within the framework of the DFT and many-body perturbation theory. The model system of InP was defined through a 2 × 2 × 2 supercell containing 64 atoms. The crystal structure optimizations were performed using the PBEsol [51]. The Brillouin zone was sampled using an 8 × 8 × 8 Γ-centered grid. The forces and total energies were converged to 0.01 eV Å^−1^, 10^−4^ eV, respectively. The crystal structures of TM-hyperdoped InP (TM = Ti, V, Cr, Mn), with general formula TM_x_In_1-x_P, were then built by replacing one In atom with a TM, which led to TM concentrations of x = 0.03, i.e., a concentration of 8.51 ×10^22^ cm^−3^. Thus, the TM concentration studied was higher than the one required to yield high absorption coefficients for new sub-bandgap absorptions and to avoid non-radiative recombination [52].

In order to get over the bandgap underestimation of standard DFT methods, the electronic structure and optical absorption properties of InP and TM@InP were calculated through the DFT+*U* approach by using a PBE (PBE+*U*) exchange–correlation functional. This formalism was satisfactorily applied to other III–V semiconductor materials [53]. DFT+*U* formalism introduces a strong intra-atomic interaction in a Hartree–Fock-like manner, wherein the strength of the intra-atomic interactions is described by *U* (on-site Coulomb) and *J* (on-site exchange) parameters. According to the formalism proposed by Dudarev et al., only effective *U_eff_ = U − J* parameters can account for the Coulomb interaction [46]. In this paper, the *U_eff_* value was tuned to resemble the bandgap value of InP, calculated using a reference method (see below). Hence, several *U_eff_* values (*U_eff_* = 5, 8, 10, 15, 18 eV, see Appendix A) for the In atom were tested. As seen in Appendix A, a value of *U_eff_* = 15 eV yielded an analogue bandgap than those calculated with the reference method, as well as in agreement with the experimental data [4]. We are aware that *U* should be chosen for each system separately. However, as seen below, *U_eff_* = 15 eV is also adequate for TM@InP compounds with respect to the benchmark method.

The many-body perturbation theory in the *GW* approach [48] was used as reference method. Concretely, *GW* calculations were carried out to correct the PBE eigenvalues without further interactions, i.e., the *G*_0_*W*_0_ approach [48], wherein the calculation starts from the DFT eigenvalues and eigenfunctions to obtain many-body *GW* self-energy. It has been successfully proven that this method yields results that are in good agreement with the experimental results for IGB materials [34,35]. The *G*_0_*W*_0_ approach was carried out with a Γ-centered 4 × 4 × 4 k-point to sample the Brillouin zone and 512 bands.

Finally, the optical properties were assessed through the absorption coefficient derived from the dielectric functions as, implemented in VASP code. The imaginary part of the dielectric function was calculated as the sum of the independent transitions between Kohn–Sham states, without local field effects, while the real part was obtained from the imaginary part by means of the Kramers–Krönig relations. The imaginary part was decomposed as over independent transitions by using a home-modified version of the original OPTICS code developed by J. Furthmueller [50].

## 3. Results

### 3.1. Crystal Structure

InP semiconductor crystallized in a symmetry group of *F-43m -T_d_^2^* (216) with a lattice constant of *a* = 5.873 Å [54]. In this structure, each In atom was surrounded by four P atoms in a tetrahedral environment and vice versa (see Figure 2). Table 1 gathers the lattice cell parameters and the most representative bond lengths for the optimized crystal structure (defined as a 2 × 2 × 2 supercell containing 64 atoms) of InP and TM@InP. As previously mentioned, in the case of the native host material, optimized structures led to lattice parameters and bond lengths that were in good agreement with the experimental data. The substitutional doping of In by a TM did not lead to important changes in the lattice cell of InP. For TM@InP compounds, TM atom was surrounded by a tetrahedron made up of four P atoms, with average bond lengths equal to *d* = 2.53 Å. This TM–P bond length was slightly lower (0.01 Å) compared with *d*(In-P) in the native compound. In addition, the six second nearest In atoms to the transition metal also defined a distorted octahedral void with a TM–TM distance equal to 4.16 Å. At the short range (i.e., in the vicinity of a TM atom, see Figure 2), In-P bond distances were barely increased with respect to the native compound (up to 0.02 Å), while no changes were found at the long range. Finally, P–P and In–In distances could be used as a measurement of the void distortion wherein TM was placed, compared with the In atom. In fact, the P–P and In–In bond lengths were 4.16 and 4.19 Å, respectively, at the short range (see *d_1_* (P–P) and *d_1_* (In–In) in Figure 2). However, P–P/In–In distances at the long range in TM@InP were slightly shorter/equal to P–P/In–In distances in the native compound. These variations were due to the smaller atomic radius of the TM compared with the In atom. Based on the results reported here, it can be concluded that the presence of TM atoms does not lead to an important crystal structure distortion in the host material.

### 3.2. Heat of Solution of TM@InP

The thermodynamic stability of the IGB materials studied in this paper were assessed through their solution enthalpies (heat of the solution, Δ*H_s_*). For TM_x_In_1-x_P, the heat of the solution can be defined as:Δ*H_s_* = *E^TM@InP^* − *E^InP^* − *μ_TM_* − *μ_In_*(1)
where *E^TM@InP^* and *E^InP^* are the total energies of TM-hyperdoped InP compounds (TM_x_In_1-x_P, x = 0.03) and the native host material (InP), respectively. *μ_TM_* and *μ_In_* are the chemical potential of the incorporated T*(*M atom (TM = Ti, V, Cr and Mn) and the In (replaced) atom, respectively. These chemical potentials are given by: *μ_x_ = μ_x_^Ref^ + Δμ_x_* (X = TM or In), where *μ_x_^Ref^* stands for the total energy of the X atom in its reference phase (see Appendix A), and Δ*μ_x_* is the change in the chemical potential of the X atom when forming the compound of interest with respect to the chemical potential of its elemental phase. In addition, the stable phase field of InP implies that:Δ*μ_In_* ≤ 0; Δ*μ_P_* ≤ 0(2)
Δ*μ_In_* + Δ*μ_P_* = Δ*H_f_(InP)*(3)
where Δ*H_f_(InP)* is the formation enthalpy of InP. Finally, the following constraints were also imposed: (i) P-rich conditions, which imply:Δ*μ_P_* = 0 → Δ*μ_In_* = Δ*H_f_(InP)*(4)

(ii) the highest TM quantity was restricted by the formation of TMP as a competing phase. This secondary phase was defined with the same structure as InP, where all In atoms were replaced by a TM. Thus, both chemical potential of TM and P atoms must satisfy:Δ*μ_TM_* + Δ*μ_P_* ≤ *ΔH_f_(TMP)*(5)

In all cases, the structure was relaxed at PBEsol level. For more details about the Δ*H_s_* calculation, see reference [55]. The calculated heat of the solution is shown in Figure 3. The values for the heat of the solution lay between −22.08 eV (TM = Cr) and 14.33 eV (Ti). The calculated heat of the solution decreased from Ti (2 *d*-electrons) to Cr (*d*-5 electrons), while substitution with Mn (*d*-5 electrons) led to a slightly larger solution heat than those obtained for TM = Cr. As mentioned, the heat of the solution is negative for most TM-hyperdoped compounds (*ΔH_s_* is positive only for TM = Ti), which means that substituting In with TM would be favored energetically. Positive *ΔH_s_* values for Ti@InP would also be related to the atomic radius. It must be noted that, among selected TM, Ti atoms own the largest atomic radius, which would lead to a less-favored In replacement. In addition, the low positive heat of the solution can be overcome through adequate experimental conditions. In fact, positive heats of solution have been calculated for other IGB materials based on III-V semiconductor materials such as Mn@GaAs and Ti@GaP [39], which were experimentally synthesized [37,56].

### 3.3. Electronic Band Structure

It is well known that sunlight absorption features are directly related to the electronic structure of the semiconductor material. Therefore, an accurate description of the electronic structure is needed. To overcome the bandgap underestimation obtained with common DFT methods, the electronic structure and optical absorption features of InP and TM@InP materials was calculated using PBE+*U*, while more accurate ab initio computational methods, such as many-body perturbation theory in the *G_0_W_0_* approach, were used as reference methods (see above) [48]. The electronic structure and partial density of states (PDOS) of the native host semiconductor (InP), calculated with PBE+*U* (*G_0_W_0_*), are shown in Figure 4 (Appendix A). The results presented for InP were in agreement with experimental data [4]. The PBE+U (*G_0_W_0_*) results provided a direct bandgap of 1.37 eV (1.37 eV) at Γ point, which agreed with the experimental value (1.35 eV). As mentioned, both PBE+U and *G_0_W_0_* approaches give similar band structure and density of states patterns. A detailed inspection of the projected density of states (Figure 4) shows that the electronic structure near the valence band maximum (VBM) was mainly formed by participation of In-5*p* and P-3*p* states, while the conduction band minimum (CBM) was mainly due to In-5*s*, In-5*p* and P-3*p* states.

Figure 5 (Appendix A) shows the electronic band structure and PDOS of TM@InP calculated with PBE*+U* (*G_0_W_0_*). Once again, both PBE+*U* and *G_0_W_0_* methods yielded a similar qualitative band structure and density of states patterns (even though calculated gaps with PBE+*U* were slightly underestimated with respect to *G_0_W_0_*). A detailed analysis of the PDOS showed that, upon In replacement by TM, the CB and VB did not change their nature significantly: the valence band maxima (VBM)was mostly due to the hybridization between In-5*p* and P-3*p* states, and the conduction band minimum (CBM) was due to In-5*s*, In-5*p* and P-3*p* states. For the spin-down channel, TM@InP compounds yielded a similar band structure pattern with a direct bandgap at Γ-point, which was opened to ~ 1.52 eV (1.63 eV), i.e., 0.15 eV (0.26 eV) with respect to InP.

For the spin-up component, when one In atom was substituted by TM, several new levels (green color in Figure 5) appeared between the edge bands of the host semiconductor (the blue and orange lines in Figure 5). Electronic structure changes upon TM hyperdoping could be described as a direct consequence of the crystal field created by P atoms around TM one. As discussed above, the TM atom was placed in a tetrahedral environment of P neighboring, with an TM–P bond length equal to 2.53 Å. In a high spin configuration, as surely was the case here, the interaction between the TM and P in a tetrahedral symmetry would cause a split of the TM-3*d* orbitals into a low-energy *e_g_* doublet (due to $d_{z^{2}}$ and $d_{x^{2}-y^{2}}$) and a high-energy *t_2g_* triplet (*d_xy_, d_xz_* and *d_yz_*), see Figure 6 [57]. This split was easily noted for all TM@InP compounds, except for TM = Ti. Starting from the Ti: 4*s*^2^3*d^2^* and In: 5*s*^2^5*p*^1^ shell configurations, the *e_g_* doublet would be semi-filled due to the extra electron, while *t_2g_* orbitals would be empty. As seen in Figure 5, the *E_Fermi_* penetrated in the first conduction band near the CBM (located at Γ-point), which was mainly due to Ti-3*d* orbitals. A detailed inspection of the PDOS showed that low-energy *e_g_* doublet comprised two flat states on the Fermi level, whilst the first conduction bands were defined by the high-energy *t_2g_* states (although they were also hybridized with higher conduction bands belonging to the InP). In short, the substitutional doping would lead to a bandgap increase up to 1.49 eV at Γ-point (1.60 eV). For Ti@InP, VB–CB energy difference due to InP host semiconductor was opened to 1.88 eV (2.25 eV).

For TM = V (4s^2^3*d^3^*), the low-energy *e_g_* doublet (fully occupied with two electrons) was located at 0.39 eV (0.51 eV) below the *E_Fermi_*. In addition, this level was well separated from the remaining valence band states belonging to the host InP semiconductor. The low-energy conduction bands were mainly due to *t_2g_* manifold, which also contributed to the higher conduction bands. At Γ-point, the energy split between *e_g_* and *t_2g_* states was 0.66 eV (1.21 eV). Once again, the bandgap of the host semiconductor was importantly opened. Though the VB–CB energy difference for those bands belonging to the host semiconductor reached a value of 1.76 eV at Γ-point (2.02 eV), the bandgap of V@InP was notably smaller (0.66 eV), which corresponded to the energy split between *e_g_* and *t_2g_* states).

For Cr@InP (Cr: 4*s^1^*3*d^5^* shell configuration), the low-energy *e_g_* doublet was full of electrons, as well as close to the conduction band edge of the host material. In addition, Cr 3*d*-occupied states also contributed to lower conduction bands. As seen in Figure 5, there was an energy difference of 0.28 eV. The high-energy *t_2g_* manifold met the requirements to be defined as IGB: (i) the band was partially filled (the Fermi level crossed it); (ii) the IGB was well isolated from both the CB (with an energy difference equal to 0.34 eV (0.49 eV) at Γ-point) and VB (the energy difference between the IGB and the VB belonging to InP host material *e_g_* states was 1.28 eV (1.24 eV)); (iii) it had a small dispersion (with the largest bandwidth = 0.20 eV along the L-Γ and X-Γ’ directions), without being in a localized defect state. As seen in Figure 6b, the wavefunction of the in-gap band state overlapped with the interacting orbitals of the neighboring atoms, which pointed to a non-localized defect state. The bandgap (VB–CB energy difference) of the host material increased to 1.62 eV at Γ-point (1.74 eV), as long as the presence of the IGB (due to *t_2g_* states) also promoted new sub-bandgap optical absorptions. As previously pointed out, the upper-limit efficiency of an IGB solar cell could reach theoretical efficiencies up to 63.1%. Such upper limits are ruled by a host bandgap of 1.93 eV and a partial IGB which yields two sub-bandgaps of 0.70 and 1.23 eV, respectively [14,15,16]. However, the Cr@InP material yielded a host bandgap and sub-bandgap values that were far from those ideal values; a theoretical efficiency ~ 60% would be obtained based on its electronic structure [14,15,16]. This value still led to a considerable improvement with respect to the theoretical maximum conversion efficiency of ~ 33% based on the SQ model [13].

Mn@InP (Mn: 4*s^2^*3*d^5^* shell configuration) yielded slightly different results depending on the computational method. According to the PBE+*U* approach, the low-energy *e_g_* doublet (which was full) established the main contributions to the highest valence bands of this hyperdoped semiconductor, as well as important contributions to deeper valence bands (due to InP host material). Concerning *t_2g_* states, this high-energy triplet could be defined as an IGB band. Considering the fact that the band was located on the Fermi level (it was partially filled with two electrons), it was well differentiated from the band edges (the energy difference between the IGB and the VB/CB was 0.76/0.81 eV (0.81/1.18 eV) at Γ-point) and it also owned a small dispersion (the largest bandwidth was 0.23 eV). Therefore, new optical sub-bandgap transitions across the IGB could be expected for Mn@InP. The main differences between the *G_0_W_0_* and PBE+*U* methods were due to the dispersion of *e_g_* states. In short, according to PBE+*U*, Mn semi-filled 3*d* orbitals (due to *t_2g_* levels) were located on the Fermi energy level and were well differentiated from *e_g_* states. Meanwhile, according to *G_0_W_0_*, both *e_g_* and *t_2g_* manifolds were completely split at Γ-points. Nonetheless, the dispersion of *e_g_* states along L-Γ and Γ-X directions hindered the split between filled (*e_g_*) and semi-filled (*t_2g_*) Mn 3*d* levels.

### 3.4. Optical Absorption Features

Finally, the optical absorption features of InP and TM@InP compounds were studied through the dielectric function. From Figure 7, we can see that the measured absorption edge started at ~ 1.33 eV, which corresponds with the bandgap of the material (1.35 eV); then, the absorption spectrum showed some shoulders, and finally exponentially increased for the higher energies [58]. Although the approach that was used to compute the dielectric functions was only based on direct transitions, our results agreed well with the experimental data and reproduced the absorption behavior for InP. As matter of fact, InP is a well-known direct bandgap material. According to the DFT results, the absorption edge began at ~ 1.30 eV. The two peaks located at roughly 1.75 and 2.15 eV nicely reproduced the most abrupted shoulder in the experimental absorption spectrum at 1.75 and 2.15 eV.

Hyperdoping with TM led to an improvement in the absorption features below the bandgap of the InP. As seen in Figure 7, the absorption was extended up to 0.5 eV (as seen below, absorption peaks at lower energies did not contribute to the photocurrent generation), and there was an important increase in the photon absorption. The gained photon absorption was not only noted below the bandgap, but also for energies higher than the bandgap of InP. The highest absorption improvements were obtained for Cr@InP. Aimed at verifying the role of the TM in the absorption enhancements of TM@InP materials, the imaginary part of the dielectric functions was decomposed as a sum of the independent transitions (Figure 8). As discussed above, for the spin-down component, the main effect under TM hyperdoping was a bandgap decrease in VB and CB, mainly due to the host material, while 3*d* states were highly hybridized with the CB. Therefore, the total contribution from the spin-down channel was labeled as VB–CB. Except for Ti@InP, the absorption spectra of TM@InP (TM = V, Cr, Mn) was characterized by several peaks in the region below 1.50 eV. The first peak (0.20–0.25 eV) was due to the electronic transitions between the different states forming the high energy *t_2g_* triplet (labeled as IGB-IGB in pink color), which was defined as IGB for TM = Cr and TM = Mn. Although these low-energy transitions contributed to the overall absorption process, they were not directly related to the photocurrent generation of the device [43].

Ti@InP and V@InP showed a bandgap decrease due to new transitions where 3*d* states were involved. Although both compounds owned 3*d* states in the forbidden gap of the material, none of them met the requirements to be an IGB. Therefore, the total absorption was due to the VB–CB transitions belonging to the host material (VB–CB transitions in blue), the VB–CB transitions where 3*d* states took part as donors (for simplicity, they are labeled as IGB–CB, in a red color) or acceptor levels (VB–IGB, labeled in a green color). For Ti@InP, the absorption edge considerably increased from ~ 1.50 eV due to VB–IGB transitions, which were related to the energy differences of 1.49 eV between the valence and *e_g_* states (Figure 5). The small absorption peak at ~ 0.65 eV mainly originated from IGB–CB (in concordance with the energy difference ~ 0.70 eV that was measured for Ti@InP at L-point and along X-Γ’ directions). For V@InP, the dielectric function gradually increased from 0.70 eV due to VB–IGB, while VB–CB transitions were mainly contributors from 1.50 eV. Mn@InP was the only system that yielded different results depending on the applied method (PBE*+U* or *G_0_W_0_),* above the split energy between *e_g_* and *t_2g_* states and, consequently, on the formation of an IGB. The enhancement of the absorption between 0.50 eV and 1.50 was due to IGB–CB (it started to appear at 0.50 eV, reaching the highest intensity at 0.90 eV) and VB–IGB transitions (which began at 0.70 eV, reaching the highest intensity at 0.85 eV), respectively. For energies larger than 1.50 eV, VB–CB transitions were the main contributors to the total absorption. It should be noted that, based on the *G_0_W_0_* electronic structure*,* that VB–IGB transitions would barely contribute to the total dielectric function, decreasing the gained absorption.

Among the different hyperdoped InP materials, Cr@InP provided the best improvement of the gained photo-absorption. As seen in Figure 7 and Figure 8, the absorption was extended below the bandgap up to 0.75 eV due to VB–IGB transitions, which reached the highest intensity at 1.35 eV. There was also a small IGB–CB contribution (the highest intensity was at 1.15 eV). Both VB–IGB and IGB–CB transitions were the main contributors to the absorption of energy ~ 1.75 eV, which corresponded to the electronic transitions between the deeper valence bands and the IGB, and between the IGB and the higher conduction bands. Regarding the VB–CB transitions, they were the main contributors to absorption energies over 2.00 eV. In short, all TM@InP compounds studied in this paper provided an absorption improvement through new transitions, due to the TM 3*d* states located in the forbidden gap of the host material. However, the presence of an IGB with the adequate properties for Cr@InP yielded the highest absorption improvement in the whole range of energies with photovoltaic applications.

## 4. Conclusions

The structural, electronic and sunlight-absorption properties of transition metal-hyperdoped InP materials, i.e., TM@InP (TM = Ti, V, Cr, Mn), with the general formula TM_x_In_1-x_P (x = 0.03) wereinvestigated using ab initio methods. The equilibrium structure of these compounds was not importantly distorted due to the TM’s presence, while the energetic of the hyperdoping process was compatible with the possibility to experimentally obtain TM@InP materials under adequate conditions. The bandgap underestimation problem of the common density-functional theory methods (DFT) was bypassed by using more accurate electronic structure methods, such as many-body perturbation methods in the *G_0_W_0_* approach and DFT along with the Dudarev approach (PBE+*U*, *U_eff_* = 15 eV) to deeply assess the electronic structure of InP and TM@InP compounds. Both methods provided similar electronic structure information. The *G_0_W_0_* approach needs high computational requirements; thus, the electronic structure and optical absorption features were calculated by using DFT+*U*, while *G_0_W_0_* was used as benchmark method. For the spin-up component, the electronic structure results can be interpreted in concordance with crystal field theory, i.e., TM 3*d* states split by the near tetrahedral environments in two manifolds: a low-energy *e_g_* doublet and a high-energy *t_2g_* triplet. Briefly, TM hyperdoping leads to a bandgap decrease for TM = Ti, V, although those levels are mainly located in the forbidden gap of the host InP material. For TM = Cr, Mn (for TM = Mn, such conclusions are only obtained with PBE+*U*), the high-energy *t_2g_* triplets meet the requirements to be defined as an IGB. Thus, a theoretical efficiency ~ 60% could be obtained based on its electronic structure, which leads to a considerable improvement with respect to the theoretical maximum conversion efficiency ~ 33% based on the SQ model. Finally, the light absorption features were investigated through the imaginary part of the dielectric function. The optical absorption coefficients are greatly enhanced in the whole range of energy for photovoltaic applications and the absorption range is also largely extended up to 0.5 eV due to new electronic transitions, where TM 3*d* states take part at donor or acceptor levels. The best results are obtained for Cr@InP, for which the gained photo-absorption considerably benefits from the two photons process induced by the IGB.

## Figures and Tables

**Figure 1 nanomaterials-10-00283-f001:**
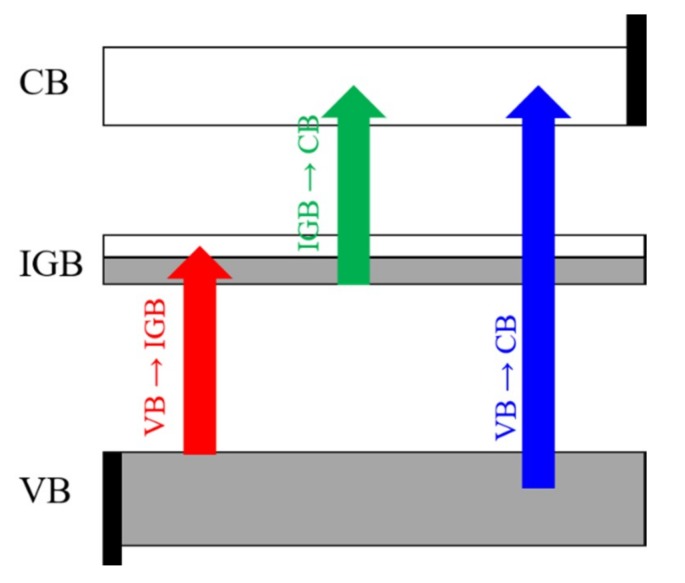
The IGB principle: photons of different energies promoted electrons VB to CB (VB→CB), as well as VB→IGB and IGB→CB, widening the photon range used.

**Figure 2 nanomaterials-10-00283-f002:**
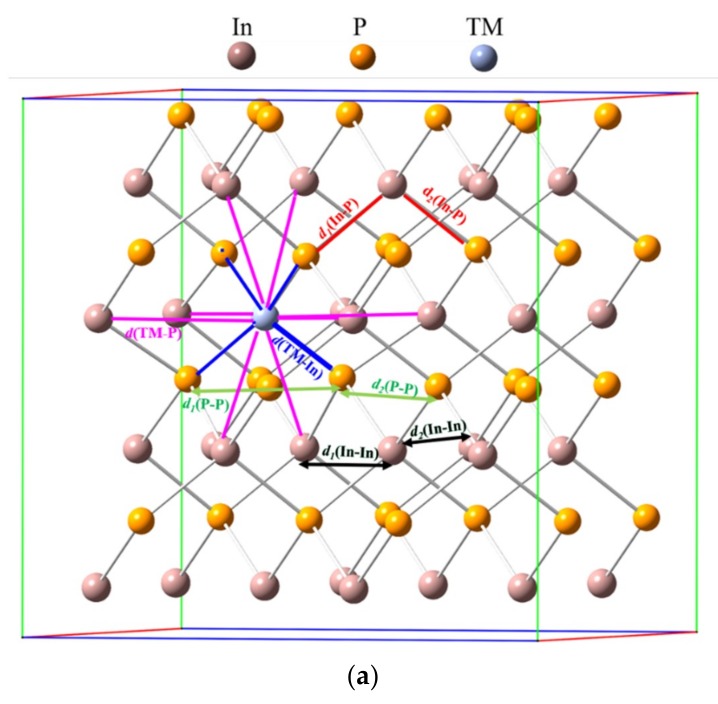
(**a**) Crystal structure of TM@InP along with main bond distances; (**b**) transition metals (TM) in a tetrahedral environment; (**c**) TM in an octahedral environment.

**Figure 3 nanomaterials-10-00283-f003:**
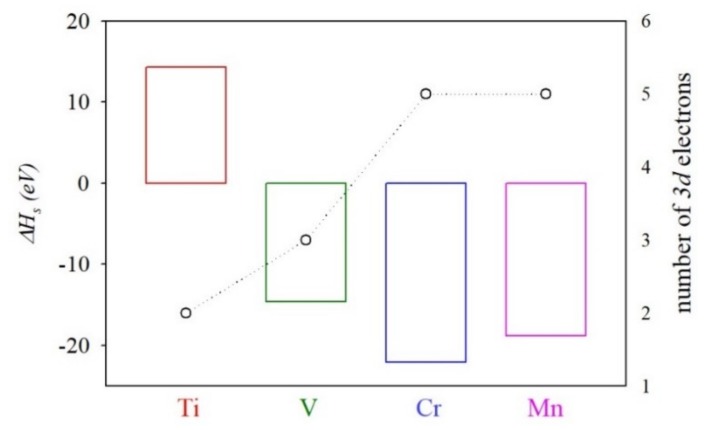
Calculated heat of solution (*ΔH_s_*) of TM@InP compounds (TM = Ti, V, Cr, Mn) along with the number of *3d* electrons.

**Figure 4 nanomaterials-10-00283-f004:**
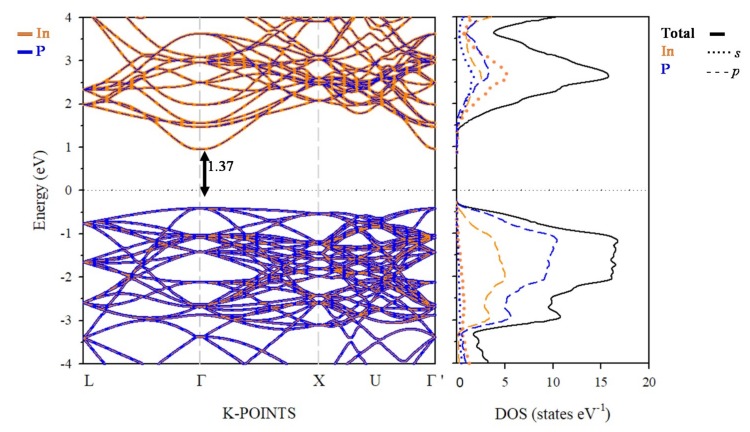
Projected electronic band structure (left) and density of states (DOS) (right) of InP calculated within PBE+U (*U_eff_* = 15 eV) approach, along with main energy differences at Γ-point. The zero energy was set at the Fermi level (black dotted line). The spin-up and spin-down bands and DOS are not given separately, as both contributions were the same for the native InP material.

**Figure 5 nanomaterials-10-00283-f005:**
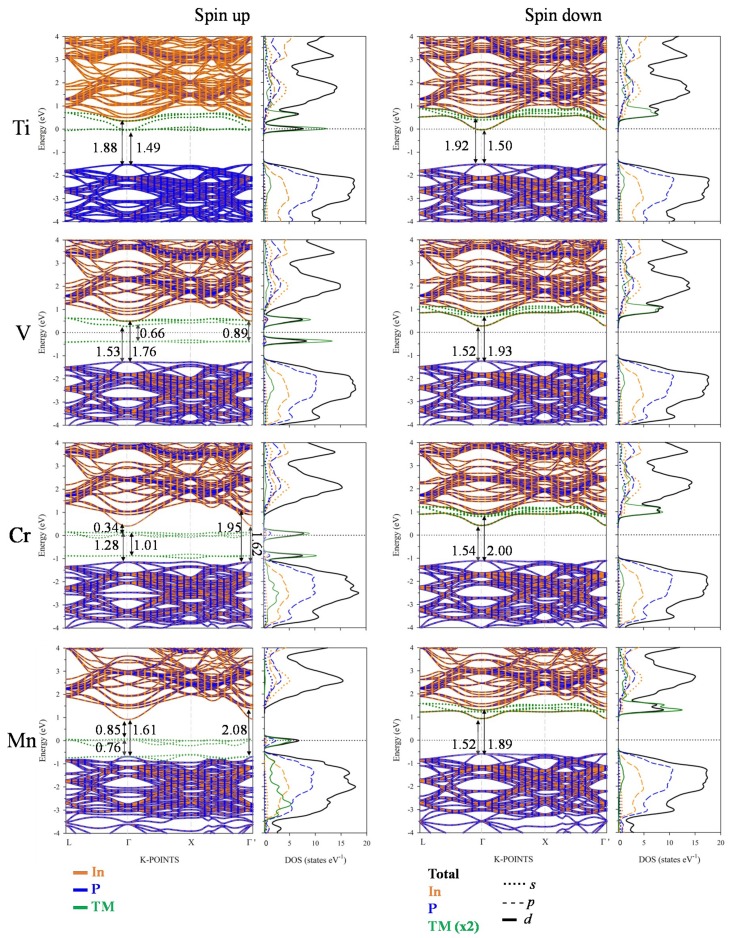
Projected electronic band structure (only the main atomic contribution was considered) and density of states of TM@InP calculated within PBE+*U* (*U_eff_* = 15 eV) approach, along with main energy differences at Γ-point. The zero energy was set at the Fermi level (black dotted line).

**Figure 6 nanomaterials-10-00283-f006:**
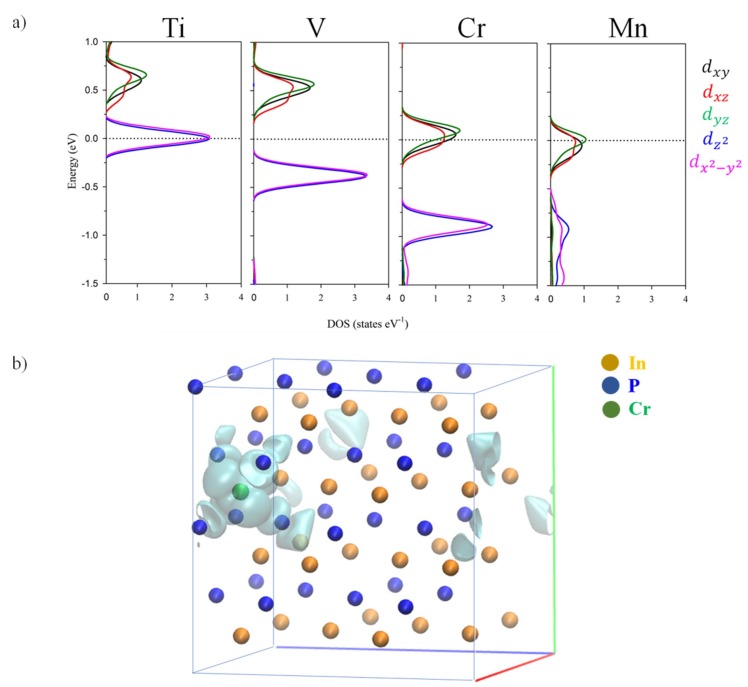
(**a**) Projected density of states of TM@InP showing the contribution of TM 3d-orbitals. The zero energy was set at the Fermi level (black dotted line); (**b**) the partial charge density of the intermediate band state for Cr@InP.

**Figure 7 nanomaterials-10-00283-f007:**
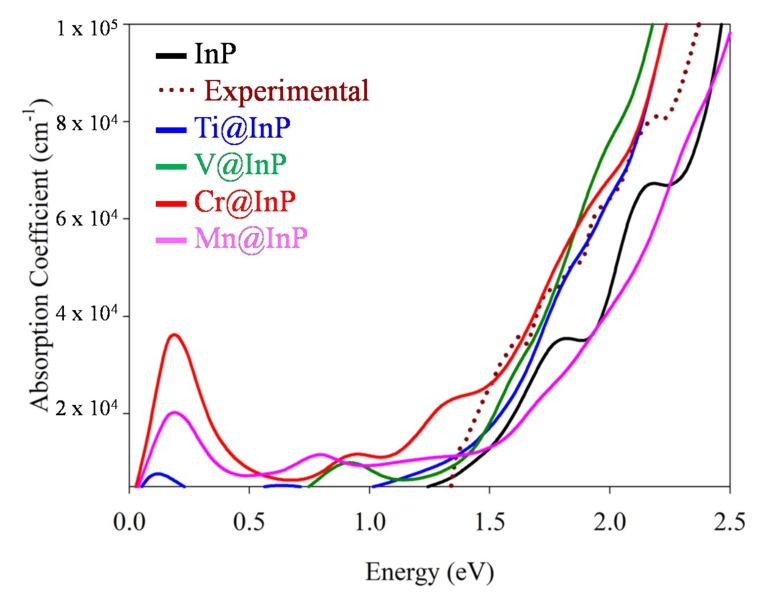
Absorption coefficients for InP and TM@InP (TM = Ti, V, Cr, Mn) compounds, along with experimental data for InP (taken from reference [58]).

**Figure 8 nanomaterials-10-00283-f008:**
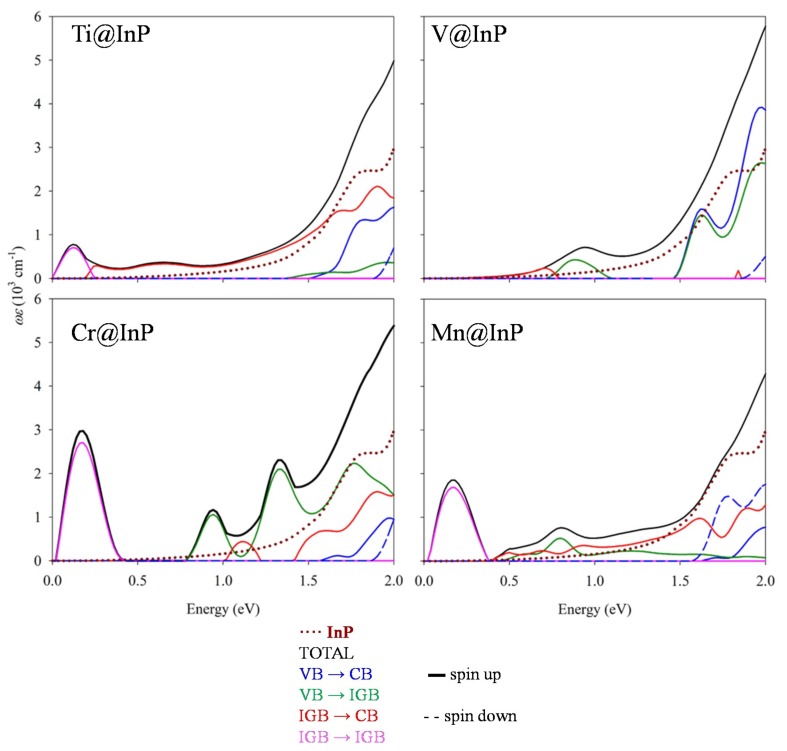
Partial contributions to the imaginary part of the dielectric function *ε*, multiplied by the frequency *ω*, for TM@InP.

**Table 1 nanomaterials-10-00283-t001:** Lattice parameters (*a*) and bond distances (*d*) obtained for optimized crystal structure of InP and TM@InP by using PBEsol functional along with experimental data of InP. For more details, see Figure 2. Units are in Angstroms (Å).

			TM@InP			
	InP *^a^*	InP	TM = Ti	TM = V	TM = Cr	TM = Mn
*a*	5.87	5.87	5.86	5.86	5.86	5.86
*d*(In-P)	2.54	2.54				
*d_1_*(In-P) *^b^*			2.55	2.56	2.56	2.56
*d_2_*(In-P) *^b^*			2.54	2.54	2.54	2.54
*d*(P–P)	4.15	4.15				
*d_1_*(P–P) *^b^*			4.16	4.16	4.16	4.16
*d_2_*(P–P) *^b^*			4.14	4.14	4.14	4.14
*d*(In–In) *^b^*	4.15	4.15				
*d_1_*(In–In) *^b^*			4.19	4.19	4.19	4.19
*d_2_*(In–In) *^b^*			4.15	4.15	4.15	4.15
*d*(TM–P)			2.53	2.53	2.53	2.53
*d*(TM–In)			4.14	4.14	4.14	4.14

*^a^* Experimental data taken from reference [54]. *^b^* The average values are collected for two kinds of bond lengths: (*d_1_*) the closest ones to TM–P bond lengths and (*d_2_*) the remaining ones.

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
