# Peer review of "Transition Metal-Hyperdoped InP Semiconductors as Efficient Solar Absorber Materials"

_nanomaterials, 2020, doi:10.3390/nano10020283_

Round 1
Reviewer 1 Report
The reviewed article is an interesting theoretical work. It would be even more interesting when comparing obtained fitted values and characteristics with experimental results for TMxIn1-xP . The practice shows that the theoretical values don’t sometimes coincide with the experimental results. I only have some questions to the authors:
Why did you choose such a molar composition of TmInP compound (TMxIn1-xP (x = 0.03))? Are TmInP compounds accessible to you? In the literature, one can find papers even from the 1990s regarding the production of InP doped Ti or Mn (e.g. grown by low pressure organometallic vapor phase epitaxy). What is the origin of the characteristic peaks on the experimental curve in fig. 6 (taken from references [58])? They are not visible in fig. 4 in [58].
Despite this, I think that the reviewed article should be published in Nanomaterials after minor corrections.
Reviewer 2 Report
Please find attached.
